# A SIMPLE AND SCALABLE SHAPE REPRESENTATION FOR 3D RECONSTRUCTION

## ABSTRACT

Deep learning applied to the reconstruction of 3D shapes has seen growing interest. A popular approach to 3D reconstruction and generation in recent years has been the CNN decoder-encoder model often applied in voxel space. However this often scales very poorly with the resolution limiting the effectiveness of these models. Several sophisticated alternatives for decoding to 3D shapes have been proposed typically relying on alternative deep learning architectures. We show however in this work that standard benchmarks in 3D reconstruction can be tackled with a surprisingly simple approach: a linear decoder obtained by principal component analysis on the signed distance transform (SDF) of the surface. This approach allows easily scaling to larger resolutions. We show in multiple experiments it is competitive with state of the art methods and also allows the decoder to be fine-tuned on the target task using a loss designed for SDF transforms, obtaining further gains.

## 1 INTRODUCTION

In recent years there has been an increased interest in extending the successes of deep learning to problems requiring analysis and representation of 3D shapes. This includes many long standing tasks such as 3D shape reconstruction (Wu et al., 2016a; Choy et al., 2016) from single or multiple views, as well as shape completion Mescheder et al. (2018). There are a number of applications of these methods in robotics, surgery, and augmented reality.

A popular approach has been the use of CNN decoder-encoder architectures Choy et al. (2016) as popularized in applications like segmentation Chen et al. (2017). Here for example in the single view reconstruction 2D CNN will encode the 2-D image and a 3D CNN decoder model will produce the final representation in voxels. This however is ineffective in larger resolutions and does not make full use of the structure of the object. Similar problems arise in more general attempts to learn latent variable models of shapes (Wu et al., 2016b; Gadelha et al., 2017). Here one may be interested in tasks such as unconditional generation and reconstruction.

More recently authors have considered alternative representations of shapes to a standard 3D discretized set of voxels(Choy et al., 2016; Girdhar et al., 2016b; Wu et al., 2016a; Yan et al., 2016; Zhu et al., 2017), one that can permit more efficient learning and generation. These include point clouds Fan et al. (2017), meshes(Wang et al., 2018; Georgia Gkioxari, 2019), and signed distance transform based representations (Michalkiewicz et al., 2019; Park et al., 2019). To date there is not an agreed upon canonical 3-D shape representation for use with deep learning models. Furthermore authors have considered alternative architectures aimed at dealing with this problem (Richter & Roth, 2018; Tatarchenko et al., 2017). However, in this work, we ask whether a simpler approach can yield strong results. Building on the recent use of the Signed Distance Function (SDF) in shape representation we demonstrate a simple latent shape representation that can be used in downstream tasks and easily decoded. More specifically, in this work, we consider a latent shape representation obtained by applying PCA on the SDF transformed shape. We show this leads to a latent shape representation that can be used directly in downstream tasks like 3D shape reconstruction from a single view and 3D shape completion from a point cloud.

Our work a) reinforces the relevance of SDF as a representation for 3D deep learning b) demonstrates that a simple representation obtained by applying PCA on the SDF transform can lead to an effective latent shape representation. This representation allows for results competitive to state-of-the-art

in standard benchmarks. Our work also suggests more complex benchmarks than those currently considered may be needed to push forward the study of learned 3D shape reconstruction.

The paper is structured as follows. In Sec. 2 we discuss the related work. We outline the basic methods used in the experiments in Sec. 3. We show extensive quantitative and experimental results comparing the eigenSDF approach to existing methods in Sec. 4.

## 2 RELATED WORK

Point cloud based representation require a tedious step of sampling points from the surface and to generate shape subsequently inferring the continuous shape from a sample of points. Meshes present a challenge in that no clear way to generate valid meshes is available. Proposals have consisted of starting with template shapes and progressively deforming them Wang et al. (2018). This however can be problematic as it never explicitly represents the shape and may suffer issues with local coherence.

Michalkiewicz et al. (2019) and Park et al. (2019) also use the SDF representation. Unlike our work Deep Level Sets still relies on a decoder encoder CNN architecture thus not removing the desired computational constraints associated with the 3D shape modeling. DeepSDF attempts to directly fit a continuous function to each shape which gives the SDF representations. Although it avoids discretization this function can lead to a complex decoder model, for example an 8 layer network is used to fit the SDF. Another recent work Mescheder et al. (2018) similar in spirit to Park et al. (2019) learns a classifier to predict whether a point is inside or outside of the boundary, using this classifier as the shape representation. Unlike our proposal these methods cannot easily learn a latent shape representation to be applied on downstream task, since the shape is represented by the weights of the classifier or regression model. On the other hand our latent representation can easily encode an unseen shape and be easily used as a prediction target for deep learning models.

Our work can also be seen as complementary to the very recent observations in Tatarchenko et al. (2019) which highlights that good 3D single view reconstruction performance can be achieved by using retrieval or clustering methods. We note however the descriptors used in that work are more complex.

PCA has been classically used to represent shapes in a variety of contexts. For example classical methods in computer vision such as the active appearance model Edwards et al. (1998) and the 3D morphable model used in face analysis Blanz et al. (1999) are based on PCA shape representations. However, these typically are applied in a different context requiring transforming the shape to a reference set of points and applying PCA on the coordinates. Leventon et al. (2002) used signed distance functions to embed 2D curves applying PCA to obtain statistical models. To the best of our knowledge it has not been combined with the SDF representing a surface in 3D. We note level set methods and the SDF have only recently been revisited as an effective representation that can be combined with 3D deep learning Michalkiewicz et al. (2019); Park et al. (2019). Moreover it is enlightening that this classic approach to shape representation can be competitive with deep learning methods on standard benchmarks.

## 3 METHODS

In this section, we review the SDF transform and describe the simple latent representation that we consider.

### 3.1 SIGNED DISTANCE FUNCTIONS

The *Signed Distance Function* (SDF) of a given closed surface $\Gamma \subset \mathbb{R}^3$ is defined by the signed distance $d : \mathbb{R}^3 \times \mathbb{R}^3 \mapsto \mathbb{R}$ from any point $x \in \mathbb{R}^3$ to the surface $\Gamma$. Given a discretized grid $\Omega$ we can represent each shape as an SDF on $\Omega$ by

$$\phi_i(x) = \pm \inf_{y \in \Gamma} d(x, y), \tag{1}$$

with the convention that $\phi_i(x)$ is positive on the interior and negative on the exterior of $\Gamma$.

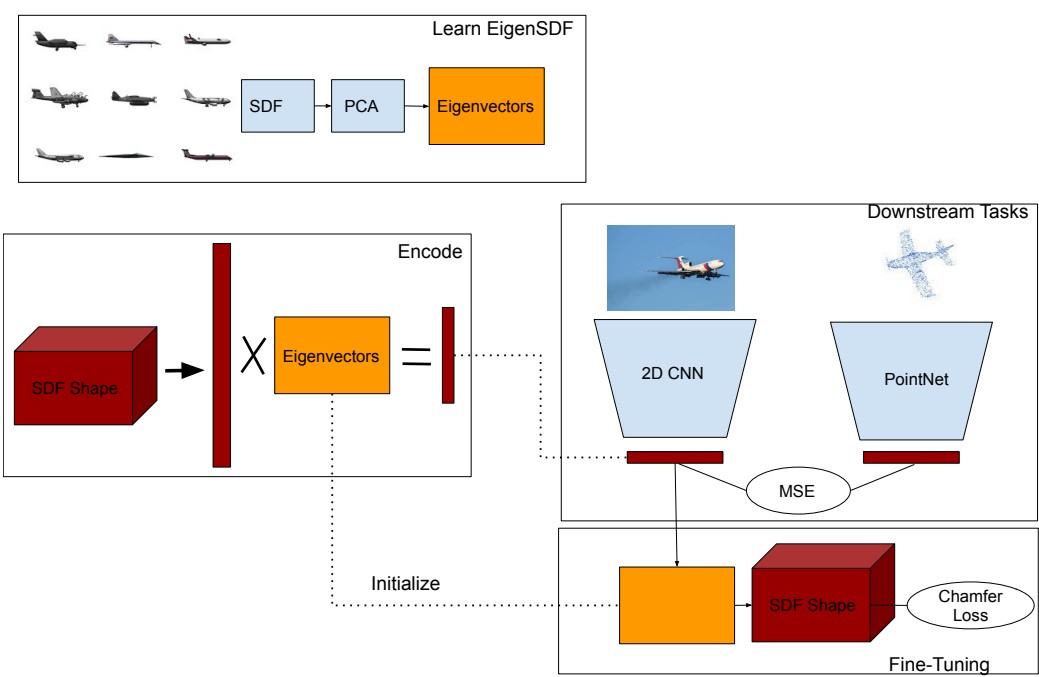

Figure 1: Overview of our experiments. We construct eigenSDF representations and evaluate them on multiple downstream tasks. As an additional approach we also consider finetuning the decoder using the eigenvectors from PCA as initializations and the chamfer loss

In Michalkiewicz et al. (2019) a CNN decoder model is used to predict the SDF representation from a latent space as well as to learn autoencoders. We note however that this representation is well structured and objects are often grouped by category, we thus ask if a much simpler linear and non-convolutional decoder model can be effective.

Michalkiewicz et al. (2019) further considers a loss function for the SDF representation that approximately minimizes the chamfer loss.

$$L_\epsilon(\theta) = \sum_{j \in D} \left( \sum_{x \in \Omega} \delta_\epsilon(\tilde{\phi}^j(x)) d^j(x)^p \right)^{1/p} + \alpha \sum_{j \in D} \sum_{x \in \Omega} (\|\nabla \tilde{\phi}^j(x)\| - 1)^2 \qquad (2)$$

with $\delta_\epsilon$ being approximated Dirac delta, $\tilde{\phi}$ inferred Signed Distance Function, and $d(x)$ the closest distance between grid point $x$ and the ground truth shape. We will further leverage this representation in the sequel.

## 3.2   EIGENSDF

We apply the PCA transform to $\mathbf{S} = \{\phi_i\}_{i=..N}$, where $N$ is the number of training examples. The eigenvectors $E$ have the shape of $(k, M^3)$ with $M$ being the grid resolution. We project each SDF to $\phi_c = \phi E^T$. Here $\phi_c$ has a shape of $(1, k)$. In the sequel we will denote this representation as the EigenSDF. Note that applying PCA to the naive voxel representation would be inappropriate as the data is binary and largely zeros. For downstream tasks we predict directly the latent representation $\phi_c$. We will also consider using $E$ as an initialization which is finetuned by training on the SDF shape representation directly using Eq. 2. A high level overview of our framework is given in Figure 1.

## 4   EXPERIMENTS

We evaluate the proposed representations on 3 tasks. These are single view 3D reconstruction, 3D reconstruction from point cloud, and 3D reconstruction with autoencoders. We take many of our

baselines and comparisons from the recently proposed Mescheder et al. (2018). These applications are evaluated on 13 categories from ShapeNet repository (Chang et al., 2015).

**Preprocessing**   In order to work on SDFs we need to have a well defined interior and exterior of an object, we have first preprocessed the meshes to make them watertight using method proposed by Stutz & Geiger (2018). Following common practice we render each ground truth mesh into 24 2D views using equally spaced azimuth angles. For each ground truth mesh we get a corresponding SDF $\phi$ of shape $128 \times 128 \times 128$.

**Metrics**   Following Mescheder et al. (2018) experimental setup, we report 3 metrics. The first one is Intersection over Union (IoU), also known as Jaccard Index, between ground truth shape $S$ and prediction $\tilde{S}$:

$$\text{IoU} = \frac{|S \cap \tilde{S}|}{|S \cup \tilde{S}|}$$

Second metric measures reconstruction error using symmetric chamfer distance:

$$\text{chamfer}(S_P, \tilde{S}_Q) = \frac{1}{2|P|} \sum_{p \in P} \min_{q \in Q} |p - q| + \frac{1}{2|Q|} \sum_{q \in Q} \min_{p \in P} |p - q|$$

Finally, we measure the angular distance using normal consistency (nc) metric:

$$\text{nc}(S_P, \tilde{S}_Q) = \frac{1}{2|P|} \sum_{p \in P} |N_{S_P}(p) \cdot N_{\tilde{S}_Q}(n_{\tilde{S}_Q}(p))| + \frac{1}{2|Q|} \sum_{q \in Q} |N_{\tilde{S}_Q}(q) \cdot N_{S_P}(n_{S_P}(q))|$$

,

where $N_S(p)$ denotes normal of point $p$ lying on surface $S$ and $n_S(q)$ denotes nearest neighbour of point $q$ lying on surface $S$.

Chamfer distance and normal consistency metrics are evaluated by sampling $10^5$ points from the surfaces, while the IoU is calculated by sampling $10^5$ points from within the bounding volume.

## 4.1   3D RECONSTRUCTION FROM SINGLE 2D VIEW

In this set of experiments, we evaluate the eigenSDF approach from Sec 3. We use a starting resolution of $128^3$. For each category we perform PCA on SDFs. We chose $k$ to capture at least 99.5% of the variance within the dataset. The image encoder is a 2D CNN whose architecture is taken from Richter & Roth (2018). We minimize the $\ell_2$ loss between $\phi_c$, the SDF projected into the latent space, and the prediction of the 2D CNN. We train this network for 100 epochs using 1e-3 learning rate, later dropping to 1e-4. Furthermore, we consider finetuning the representation starting with the eigenvectors from PCA and using Eq 2, this baseline is referred to as eigenSDF (finetuned).

In order to demonstrate that a gain is made by PCA versus just architecture we also train a linear autoencoder of the same size ($M$) and finetune it with Eq 2. This baseline is referred to as linearSDF and linearSDF(finetune).

Finally we compare to a set of standard benchmarks from the recent literature including voxel based CNN encoder/decoder Choy et al. (2016), point cloud based methods Fan et al. (2017), a mesh based method Wang et al. (2018), and the recently introduced ONet Mescheder et al. (2018).

Complete results are given in Table 1. First we observe that the simply using a same sized linear model linearSDF (finetune) is outperformed by using the eigenSDF. Compared to alternatives our method gives more significant gains in chamfer metric than all competitors and can be further improved with the finetuning. We also observe improvements in the normal consistency metric. For the IoU metric we observe that eigenSDF outperforms all methods except Mescheder et al. (2018). Note that according to Sun et al. (2018) chamfer distance is a far better metric for shape comparison than IoU.

Table 1: Single-view 3D reconstruction results on ShapeNet. We observe that compared to other state-of-the-art learning based methods eigenSDF outperforms in normal consistency and Chamfer distance. Finetuning can further improve this result. Compare to training a linear autoencoder or just finetuning the performance is substantially better, showing that eigen decomposition obtains best results.

| Chamfer↓ | | | | | | | | |
|---|---|---|---|---|---|---|---|---|
| Cat name | 3D R2N2 | PSGN | Pix2Mesh | AtlasNet | ONet | linearSDF | linearSDF (ft) | eigenSDF | eigenSDF (ft) |
| airplane | 0.227 | 0.137 | 0.187 | 0.104 | 0.147 | 0.256 | 0.250 | 0.102 | **0.080** |
| bench | 0.194 | 0.181 | 0.201 | 0.138 | 0.155 | 0.247 | 0.241 | 0.085 | **0.072** |
| cabinet | 0.217 | 0.215 | 0.196 | 0.175 | 0.167 | 0.223 | 0.220 | 0.082 | **0.065** |
| car | 0.213 | 0.169 | 0.180 | 0.141 | 0.159 | 0.232 | 0.231 | 0.072 | **0.057** |
| chair | 0.270 | 0.247 | 0.265 | 0.209 | 0.228 | 0.265 | 0.261 | 0.110 | **0.092** |
| display | 0.314 | 0.284 | 0.239 | 0.198 | 0.278 | 0.286 | 0.258 | 0.115 | **0.115** |
| lamp | 0.778 | 0.314 | 0.308 | 0.305 | 0.479 | 0.633 | 0.625 | 0.465 | **0.377** |
| loudspeaker | 0.318 | 0.316 | 0.285 | 0.245 | 0.300 | 0.262 | 0.257 | 0.097 | **0.092** |
| rifle | 0.183 | 0.134 | 0.164 | 0.115 | 0.141 | 0.282 | 0.267 | 0.148 | **0.140** |
| sofa | 0.229 | 0.224 | 0.212 | 0.177 | 0.194 | 0.273 | 0.266 | 0.192 | **0.137** |
| table | 0.239 | 0.222 | 0.218 | 0.190 | 0.189 | 0.201 | 0.195 | 0.127 | **0.115** |
| telephone | 0.195 | 0.161 | 0.149 | 0.128 | 0.140 | 0.181 | 0.177 | 0.047 | **0.045** |
| vessel | 0.238 | 0.188 | 0.212 | 0.151 | 0.218 | 0.424 | 0.440 | 0.347 | **0.335** |
| mean | 0.278 | 0.215 | 0.216 | 0.175 | 0.215 | 0.279 | 0.273 | 0.163 | **0.142** |

| IoU↑ | | | | | | | | |
|---|---|---|---|---|---|---|---|---|
| Cat name | 3D R2N2 | PSGN | Pix2Mesh | AtlasNet | ONet | linearSDF | linearSDF (ft) | eigenSDF | eigenSDF (ft) |
| airplane | 0.426 | - | 0.420 | - | **0.571** | 0.425 | 0.433 | 0.465 | 0.487 |
| bench | 0.373 | - | 0.323 | - | **0.485** | 0.372 | 0.375 | 0.382 | 0.405 |
| cabinet | 0.667 | - | 0.664 | - | **0.733** | 0.658 | 0.661 | 0.670 | 0.686 |
| car | 0.661 | - | 0.552 | - | **0.737** | 0.665 | 0.669 | 0.672 | 0.704 |
| chair | 0.439 | - | 0.396 | - | **0.501** | 0.385 | 0.401 | 0.402 | 0.414 |
| display | 0.440 | - | **0.490** | - | 0.471 | 0.388 | 0.395 | 0.406 | 0.424 |
| lamp | 0.281 | - | 0.323 | - | **0.371** | 0.201 | 0.211 | 0.230 | 0.280 |
| loudspeaker | 0.611 | - | 0.599 | - | **0.647** | 0.554 | 0.561 | 0.599 | 0.609 |
| rifle | 0.375 | - | 0.402 | - | **0.474** | 0.256 | 0.261 | 0.289 | 0.299 |
| sofa | 0.626 | - | 0.613 | - | **0.680** | 0.611 | 0.619 | 0.633 | 0.639 |
| table | 0.420 | - | 0.395 | - | **0.506** | 0.401 | 0.406 | 0.422 | 0.426 |
| telephone | 0.611 | - | 0.661 | - | **0.720** | 0.590 | 0.615 | 0.685 | 0.715 |
| vessel | 0.482 | - | 0.397 | - | **0.530** | 0.443 | 0.449 | 0.494 | 0.508 |
| mean | 0.493 | - | 0.480 | - | 0.571 | 0.457 | 0.465 | 0.488 | 0.507 |

| Normal Consistency↑ | | | | | | | | |
|---|---|---|---|---|---|---|---|---|
| Cat name | 3D R2N2 | PSGN | Pix2Mesh | AtlasNet | ONet | linearSDF | linearSDF (ft) | eigenSDF | eigenSDF (ft) |
| airplane | 0.629 | - | 0.759 | 0.836 | **0.840** | 0.713 | 0.717 | 0.809 | 0.813 |
| bench | 0.678 | - | 0.732 | 0.779 | 0.813 | 0.754 | 0.762 | 0.815 | **0.832** |
| cabinet | 0.782 | - | 0.834 | 0.850 | 0.879 | 0.771 | 0.773 | 0.880 | **0.882** |
| car | 0.714 | - | 0.756 | 0.836 | 0.852 | 0.787 | 0.802 | 0.855 | **0.878** |
| chair | 0.663 | - | 0.746 | 0.791 | 0.823 | 0.757 | 0.777 | 0.824 | **0.828** |
| display | 0.720 | - | 0.830 | 0.858 | 0.854 | 0.755 | 0.787 | 0.859 | **0.859** |
| lamp | 0.560 | - | 0.666 | 0.694 | 0.731 | 0.575 | 0.584 | 0.741 | **0.775** |
| loudspeaker | 0.711 | - | 0.782 | 0.825 | 0.832 | 0.745 | 0.755 | 0.839 | **0.846** |
| rifle | 0.670 | - | 0.718 | 0.725 | 0.766 | 0.668 | 0.695 | 0.768 | **0.793** |
| sofa | 0.731 | - | 0.820 | 0.840 | **0.863** | 0.733 | 0.739 | 0.861 | **0.863** |
| table | 0.732 | - | 0.784 | 0.832 | **0.858** | 0.731 | 0.735 | 0.801 | 0.808 |
| telephone | 0.817 | - | 0.907 | 0.923 | 0.935 | 0.851 | 0.855 | 0.938 | **0.940** |
| vessel | 0.629 | - | 0.699 | 0.756 | 0.794 | 0.704 | 0.735 | 0.802 | **0.811** |
| mean | 0.695 | - | 0.772 | 0.811 | 0.834 | 0.734 | 0.747 | 0.830 | **0.840** |

## 4.2 3D SHAPE COMPLETION FROM POINT CLOUDS

We next consider shape completion from a point cloud. This task has been considered in e.g. Mescheder et al. (2018). Similar to experimental setup of Mescheder et al. (2018) we use 13 categories from ShapeNet repository and we preprocess the meshes to make them watertight. We randomly sample 300 points from ground truth meshes and add a Gaussian noise with 0 mean and 0.05 std. We have used the same metrics as in section 4.1.

We have used PointNet encoder with a bottleneck dimension of 512 Qi et al. (2017) and linear decoder from section 4.1. We use a similar set of baselines as in the previous section and compare to eigenSDF. We observe similar large gains in the chamfer metric and competitive performance in other metrics. Results are shown in Table 2.

We can see that, similar to 3D reconstruction task, our performance in chamfer is much better, normal consistency is similar and iou is worse but still competitive.

Table 2: Results on 3D Shape completion

|  | IoU↑ | Chamfer↓ | nc ↑ |
|---|---|---|---|
| eigenSDF (ours) | 0.571 | **0.073** | 0.853 |
| 3D-R2N2 (Choy et al. (2016)) | 0.565 | 0.169 | 0.719 |
| PSGN (Fan et al. (2017)) | - | 0.144 | - |
| DMC (Liao et al. (2018)) | 0.674 | 0.117 | 0.848 |
| ONet (Mescheder et al. (2018)) | **0.778** | 0.079 | **0.895** |

Table 3: We compare eigenSDF to other methods in terms of reconstruction. We find that compared to linear autoencoders trained on voxels or SDFs, it performs better. We also compare to more sophisticated methods.

| method | IoU↑ | Chamfer↓ | NC↑ | F-score↑ |
|---|---|---|---|---|
| eigenSDF | 0.746 | 0.0425 | 0.869 | 0.484 |
| eigenSDF (ft) | **0.758** | **0.0325** | **0.896** | **0.529** |
| Linear ($\phi$) (chamfer) | 0.582 | 0.050 | 0.773 | 0.315 |
| Linear(voxels) | 0.637 | 0.067 | 0.737 | 0.384 |
| DLS (Michalkiewicz et al. (2019)) | 0.681 | 0.047 | 0.858 | 0.103 |
| TL (Girdhar et al. (2016b)) | 0.656 | 0.082 | 0.847 | 0.081 |

## 4.3 3D RECONSTRUCTION FROM LATENTS

Finally we consider a simple 3D reconstruction task Girdhar et al. (2016a). This can also be viewed as measuring the representational power of the model Mescheder et al. (2018). This evaluates reconstruction quality of eigenSDF versus other methods particularly CNN based autoencoders. The goal is to reconstruct test set shapes. We use an initial resolution of $128^3$ and reduce it to $k = 512$ as done in other works Mescheder et al. (2018). We use *cars* category from ShapeNet repository and evaluate reconstruction on unseen data. For the evaluations in addition to the metrics used in section 4.1 we further analyse the decoders using F-score Knapitsch et al. (2017).

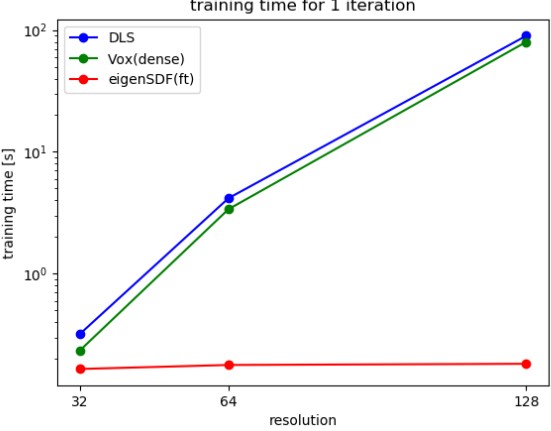

Figure 2: Training time of eigenSDF (finetuned) and dense convnets (Voxels, DLS). Figure shows on a logscale amount of seconds a network needs for forward and backward pass of 1 iteration using a batchsize of 32. Resolution 256 not shown due to clarity.

| category | DLS | | | | eigenSDF(ft) | | | |
|---|---|---|---|---|---|---|---|---|
| | IoU↑ | Chamf↓ | NC↑ | F-score↑ | IoU↑ | Chamf↓ | NC↑ | F-score↑ |
| cars | 0.784 | 0.055 | 0.804 | 0.148 | **0.821** | **0.040** | **0.909** | **0.432** |
| chairs | 0.434 | 0.360 | 0.743 | 0.066 | **0.553** | **0.125** | **0.820** | **0.168** |
| sofas | 0.581 | 0.132 | 0.779 | 0.089 | **0.647** | **0.082** | **0.867** | **0.248** |

Table 4: Comparison to the SDF based method Michalkiewicz et al. (2019) in single view reconstruction. There is marked improvement due to the ability to model higher resolution.

## 4.4 COMPARISON WITH DEEP LEVEL SETS

In this section, we compare our method to the other recent approach relying on the signed distance transform Michalkiewicz et al. (2019) and learning with the chamfer loss $L_\epsilon$. This one however uses the CNN decoder model and does not learn a latent shape representation. We have chosen a similar experimental setup of 3 subsets each having 2 000 examples from ShapeNet repository: *cars*, *sofas*, *chairs*. We observed that remaining 2 categories, *bottles* and *phones*, are too simple to see a difference in higher resolution. Reported metrics might differ due to different pre-processing techniques. Networks here are the one used in Michalkiewicz et al. (2019) paper and the one used in Section 4.1. We first compare the training time for both methods for various resolutions. Figure 2 shows the result demonstrating that it is not feasible to use the CNN decoder in higher resolutions. It is consistent with the findings of (Richter & Roth, 2018).

## 4.5 RECONSTRUCTION AND GENERATION

Finally we evaluate the performance on the single view reconstruction qualitatively. In Table 4 we can see that reconstructions (on unseen data) can be effective capturing more complex structures ignored by Michalkiewicz et al. (2019). Multiple authors have also consider generating unconditionally shapes, typically using sophisticated non-linear deep learning models like GANs and VAEs. We compare some of these to sampling a gaussian in the latent space of the eigenSDF, some qualitative results are shown in Figure 3. We observe this simple approach yields some shapes comparable to those o the complex non-linear models.

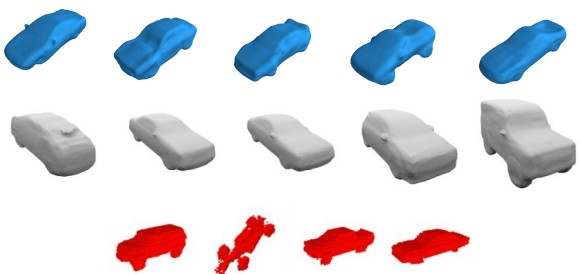

Figure 3: We compare unconditional generations of cars category. Generations from a gaussian fit to eigenSDF is shown in the top row (blue). Second row are generations from Mescheder et al. (2018) and the third row is from a 3D GAN Wu et al. (2016b)

## 5 CONCLUSION

We have shown that using a simple linear decoder coupled with the SDF representation yields competitive results. The SDF lends itself to the application of PCA yielding a strong but simple baseline for future work in learned 3D shape analysis. Moreover, our work suggests that more complex baseline datasets may be needed to further evaluate deep learning methods on 3D shape inference.

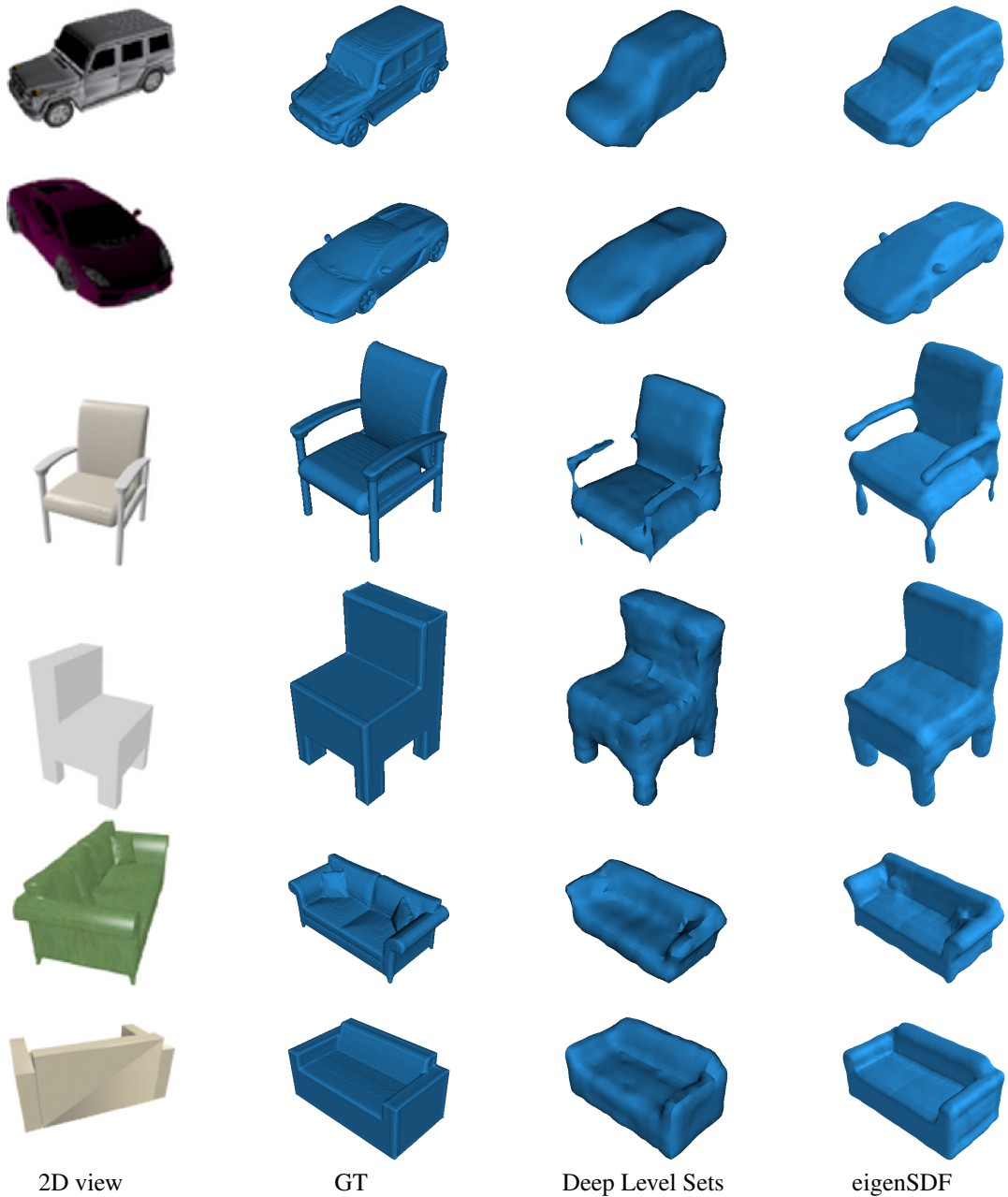

|              |     |                 |          |
| ------------ | --- | --------------- | -------- |
| 2D view      | GT  | Deep Level Sets | eigenSDF |

Table 5: We compare reconstruction of eigenSDF and the CNN decoder based Michalkiewicz et al. (2019) (which also uses SDF representations). eigenSDF permits us to operate at a higher resolution and generally produces more locally coherent results.

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

# A APPENDIX

## A.1 NUMBER OF EIGENVECTORS

In Figure 4, we show the number of eigenvectors which captures 99.5% of variance within the dataset of approximately 6000 examples of *ShapeNetCars*. Number of eigenvectors required to represent shapes from other categories vary from 512 to 2048, shown in Figures 5-6.

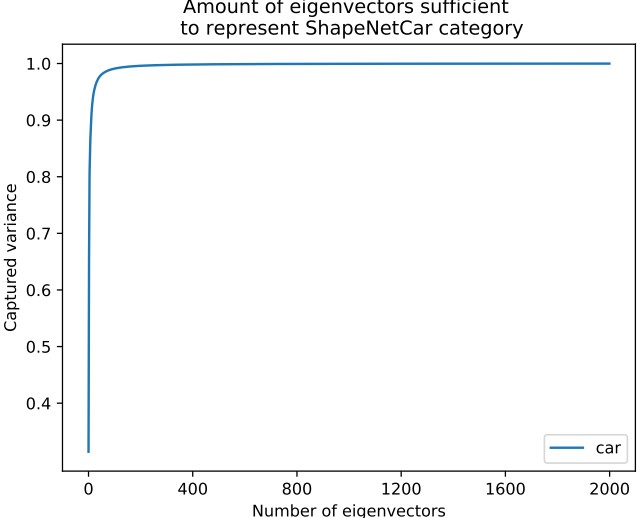

Figure 4: Cumulative fraction of total variance captured by eigenvectors obtained from applying PCA on *ShapeNetCars* dataset.

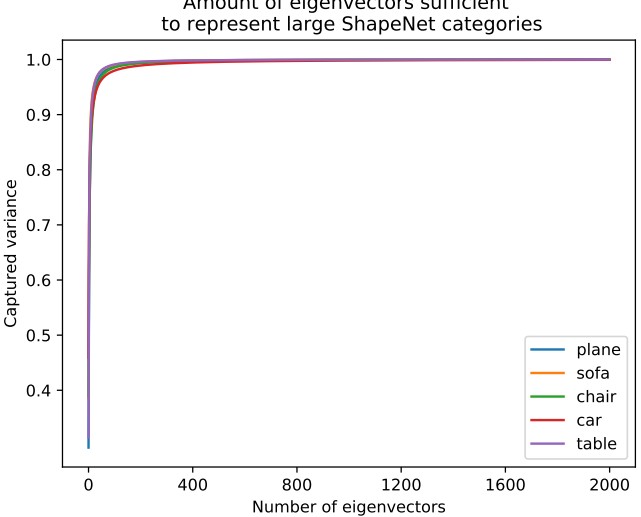

Figure 5: Cumulative fraction of total variance captured by eigenvectors obtained from applying PCA on large *ShapeNet* categories.

## A.2 PCA APPLIED TO OTHER SHAPE REPRESENTATIONS

As mentioned in the Section 1, there are 4 commonly used shape representations: voxels, Signed Distance Functions (SDF), point clouds and meshes. Because of lack of canonical order or representation it is hard to apply PCA on point clouds or meshes in a straightforward way. Compared to

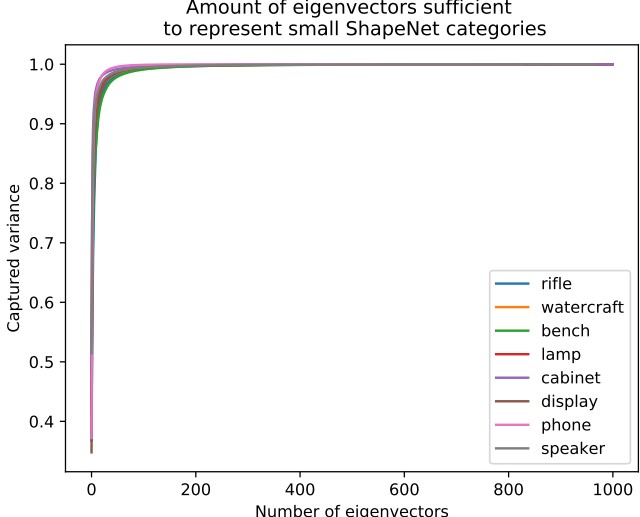

Figure 6: Cumulative fraction of total variance captured by eigenvectors obtained from applying PCA on small *ShapeNet* categories.

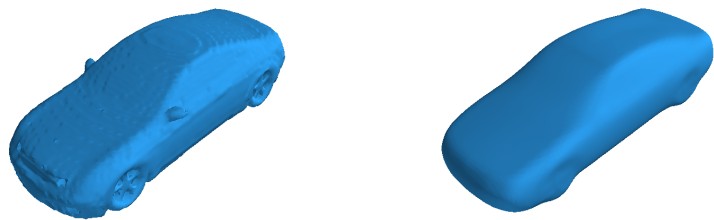

Figure 7: SDF-based pca reconstruction of an example from *ShapeNetCars* dataset. Left - Level set of a SDF of a car with resolution $128^3$, right - pca-based recostruction using 2048 eigenvectors.

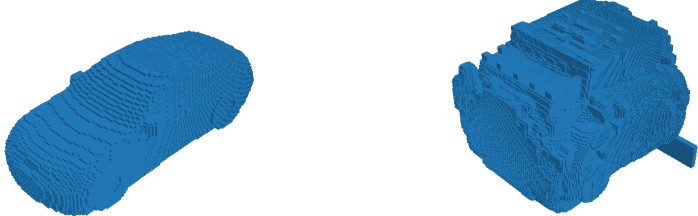

Figure 8: Failed attempt of reconstructing an example from *ShapeNetCars* dataset using voxel-based representation. Left - voxelized car with resolution $128^3$, right - pca-based recostruction using 2048 eigenvectors

the results of pca reconstruction using Signed Distance Function as shape representation, applying PCA directly on voxels yeilds poor results, as shown in Figures 7-8.

