# OpenReview forum: "A Simple and Scalable Shape Representation for 3D Reconstruction"
_ICLR.cc/2020/Conference — Reject_

### Official Review · AnonReviewer1 · 2019-10-22
**Official Blind Review #1**

**Rating:** 3

**Review:**

The paper introduces a 3D object reconstruction/completion algorithm that utilizes a simple decoder from features generated using PCA of SDF. The approach was tested in a few experiments on public benchmarks and achieves competitive results.

The overall presentation of the paper is decent, and the network structure of the proposed approach is reasonable. It's interesting to see that using a simple PCA can help improve performance using a simple network structure. The experimental results make sense, and it's nice to see the performance is reasonable as well.

I have a few questions regarding the paper:
- Without looking at the code, I don't think I fully understand the formulation of the network structure just by reading the text. For example, what is the dashed line mean in Fig. 1? What's the meaning of the dot over "E" in Sec 3.2, is it "derivatives"? If so, why not use this symbol in Eq. (2) as well? In general, I find it's a bit hard to follow when only 2-3 paragraphs are used for describing the proposed approach. It'll be good if the authors can elaborate on the approach in a more thorough manner.
- Regarding experimental results, it's nice to see that eigenSDF is better than linearSDF, demonstrating that the approach is quite effective. However it is not always the best in several metrics (as indicated in experimental result tables). I wonder if authors can provide more analysis or discussions on why this could happen, either the metric may not make too much sense in their setting, or if there is potential room for improvement. A few failure case visualizations could also be helpful in understanding the issues of the proposed approach.
- Moreover, do authors have thoughts on eigenSDF vs deepSDF (cited in the paper, published in CVPR 2019)? It'll be interesting to compare those as well, as deepSDF has proven useful in a few papers already.

**Experience Assessment:**

I have published one or two papers in this area.

**Review Assessment: Checking Correctness Of Derivations And Theory:**

I carefully checked the derivations and theory.

**Review Assessment: Checking Correctness Of Experiments:**

I carefully checked the experiments.

**Review Assessment: Thoroughness In Paper Reading:**

I read the paper at least twice and used my best judgement in assessing the paper.

---

> ### Author Response · Authors · 2019-11-13
> **Response to review #1**
>
> Thank you for your review.
>
> "For example, what is the dashed line mean in Fig. 1?"
>
> The dashed lines indicate that the same values are used in the downstream task. E.g. the encoded PCA representation is used in the downstream tasks combined with MSE. Similarly for the fine tuned version of eigenSDF the eigenvectors are used to initialize a decoder model, which is then further finetuned with chamfer loss.
>
> "What's the meaning of the dot over "E" in Sec 3.2, is it "derivatives"? If so, why not use this symbol in Eq. (2) as well?"
>
> The dot over E was a typo. E is just the matrix of eigenvalues.  Equation (2) refers to the chamfer loss, which we emphasize is only used in our models denoted eigenSDF (finetuned). Typically we just minimize the loss in the latent space of the PCA.
>
> "In general, I find it's a bit hard to follow when only 2-3 paragraphs are used for describing the proposed approach. It'll be good if the authors can elaborate on the approach in a more thorough manner. "
>
> We have updated section 3 to attempt to make it more clear. However the concept of proposed eigenSDF is simple: we learn a PCA model for the shape using the SDF representation, obtaining a simple latent shape representation. For downstream tasks (e.g. 2D image -> 3D shape) we simply minimize the MSE in the latent space, saving completely the decoding step during training, and having a very light decoder for inference.
>
> We can also finetune this entire model using the chamfer loss applied at the level of the SDF representation, but note even without this performance is already competitive.
>
> We do emphasize this high level view in several places besides the more formal Section 3 (e.g. in the introduction and in Fig 1).
>
> "Regarding experimental results, (...). However it is not always the best in several metrics (as indicated in experimental result tables). I wonder if authors can provide more analysis or discussions on why this could happen, either the metric may not make too much sense in their setting, or if there is potential room for improvement."
>
> Regarding further insights into the metrics. We want to note that Occupancy networks were explicitly trained for the IoU. On the other hand Chamfer distance is a much better metric for this task as mentioned in [1] and [2].
>
> We also want to note that single instance where LinearSDF is better than EigenSDF was actually due to a typographical error from transferring numbers to the table. This was only for chamfer distance and rifle category. Note that for rifles, the IoU and Normal Consistency measures were better for EigenSDF. We have now fixed this.
>
> "A few failure case...  of the proposed approach."
>
> One of the issues can be seen when looking at the quantitative table: similar to 3D-R2N2 or Deep Level Sets, eigenSDF is struggling with reconstructing thin objects such as examples from lamp category.
>
> A possible improvement can be the following:
>     1) pre-processing SDFs by adding a small epsilon to make the SDFs “fatter”
>     2) Training eigenSDF to learn “fat” version of examples
>     3) Switch the L2 loss to chamfer loss and the ground truth to original, thin examples.
>
> Note that levelset methods are often susceptible to good initialization procedure.
>
> "Compare against DeepSDF"
>
> First of all we want to emphasize one of the goals of our work is to show a simple (linear) baseline can be competitive to the current state-of-the-art methods on the standard tasks. Indeed DeepSDF is an interesting and related work. As discussed in the Introduction DeepSDF avoids discretization but can lead to a complex decoder model, for example an 8 layer network is used to fit the SDF.  In our case the representation is given by a simple linear transformation.  Note DeepSDF does not give a task agnostic latent variable model as in our case (aka to represent a specific shape you need to fit a separate deep NN for each shape or do it implicitly conditioned on an image for a given task). In our formulation the shape representation has an explicit small latent code and this thus allows us to perform training in latent space. We also note DeepSDF subsamples 16384 SDF  points. In our case, we capture over 99% of variance of over 2 million points. We noted that the dataset and preprocessing used in DeepSDF is different than in our work and the others we compare to thus it is difficult for us to compare at this time.  Specifically to use SDFs we need watertight meshes.
>
> [1] Sun, Xingyuan, et al. "Pix3d: Dataset and methods for single-image 3d shape modeling." Proceedings of the IEEE Conference on Computer Vision and Pattern Recognition. 2018.
> [2] Tatarchenko, Maxim, et al. "What Do Single-view 3D Reconstruction Networks Learn?." Proceedings of the IEEE Conference on Computer Vision and Pattern Recognition. 2019.
> [3] Michalkiewicz, Mateusz, et al. "Deep Level Sets: Implicit Surface Representations for 3D Shape Inference." arXiv preprint arXiv:1901.06802 (2019).

---

### Official Review · AnonReviewer3 · 2019-10-27
**Official Blind Review #3**

**Rating:** 6

**Review:**

Thank the authors for the response. I am still in favor of the idea -- applying simple, old-school method into a new problem, and I also agree with R1 and R2 that the paper is currently lack of details and experimental results. I will keep my score, but would not fight for the acceptance if R1 and R2 insist.
----------------------------------------
Summary
This paper presents a new method for 3D shape reconstruction, based on SDF (Signed Distance Function) and PCA. The basic idea is to conduct PCA on all the shapes with SDF as feature, and encode a shape by the eigenvectors from PCA. The authors present experiments on 3D reconstruction from 2D view and point clouds, which demonstrate the effectiveness of the proposed method. I lean to vote for accepting this paper since the idea is simple but novel, and it achieves good performance.
Strengths
- The idea itself is simple and novel. The basic idea of this approach is simple -- keep most information / variance by using PCA, and it is also very novel, since I have not seen papers using PCA to encode 3D shapes.
- The idea is effective. As the authors demonstrated in section 4, this approach works well, and it outperforms all other methods by a large margin according to Chamfer distance. This is impressive since such a simple method can improve the performance this much.
Weaknesses
- More analysis could be provided about how do the authors choose SDF. Choosing SDF here is obviously a reasonable choice, but is it the best? More analysis could be provided, or more experiments could be included.
Possible Improvements
As mentioned above, more analysis about why choosing SDF or more experiments about comparing SDF to other representations under this PCA approach could be provided.

**Experience Assessment:**

I have published one or two papers in this area.

**Review Assessment: Checking Correctness Of Derivations And Theory:**

I assessed the sensibility of the derivations and theory.

**Review Assessment: Checking Correctness Of Experiments:**

I assessed the sensibility of the experiments.

**Review Assessment: Thoroughness In Paper Reading:**

I read the paper thoroughly.

---

> ### Author Response · Authors · 2019-11-13
> **Response to review #3**
>
> Thank you for your review.
>
> Regarding the motivation of the PCA + SDF.  The goal of our paper was to determine if a simple latent variable model can be used to replace the typically complex decoder. PCA seems a natural choice for this however the representation it is applied to is less obvious. As noted in [1], and in many other papers, there are currently 4 main shape representations: voxels, SDFs, point clouds and meshes. Applying the PCA to voxels is somewhat inappropriate, they are binary while PCA is designed for continuous variables. We do however evaluate this now in Appendix A2, where we show 3D reconstruction with 2048 eigenvectors for a random example of ShapeNetCars using voxel-based representation and SDFs. Voxel-based reconstructions perform so poorly that we did not even consider them for quantitative evaluation.
>
> Applying PCA to point clouds and meshes is not evident. Point clouds do not have a natural ordering thus it is unclear how one can apply it here. Similarly meshes do not have any canonical representation that can be used to represent them.  We have added this discussion to Appendix A2.
>
> [1] Michalkiewicz, Mateusz, et al. "Deep Level Sets: Implicit Surface Representations for 3D Shape Inference." arXiv preprint arXiv:1901.06802 (2019).

---

### Official Review · AnonReviewer2 · 2019-11-03
**Official Blind Review #2**

**Rating:** 3

**Review:**

This paper studies the problem of learning the feature representation for predicting the 3D shape of objects, from a single image or a point cloud. The proposed approach performs PCA on the SDF field. And then the transformed feature map is learned and used as input to task-specific decoders for 3D shape prediction. The authors claims that this approach trains faster and is easier to scale, while showing competitive performance compared to state-of-the-art methods.

I am leaning towards Weak Reject. The paper is generally easy to read, but with some details missing. And I found the discussion of the results to be insufficient. I think it can be an above-threshold paper if questions are addressed during rebuttal.

Being able to easily scale to higher resolutions is claimed to be one of the main advantages, but I am not convinced that this is useful under this setting. If I understand this correctly, the number of eigenvectors k is fixed, and projecting the SDF field to this space would remove the higher frequency components of the shape. So wouldn't the number of eigenvectors be the bottleneck in representational precision, not the resolution of the output space?

What is the chosen k (number of eigenvectors)? It says k was "chosen to capture 99.5% of the variance within the dataset", but I could not find how exactly it was chosen and what value of k was used (I apologize if I missed).

Also, I think the PCA is category-specific (page 4, section 4.1). Is k dependent on the category or is it the same across all categories? Some of the other methods (if not all) used for comparison are not category-specific, so if this is true, I think the comparison may not be entirely fair and it should be made clearer.

I think the writing could be polished as well, some minor typos:

Page 2:  3rd and 4th paragraphs: continous
Page 2: anlaysis, enlightning
Page3: under eigenSDF: reprsentation
Page 4: section 4.1: refered, signficant
Page 5: “seciton”
Page 7: tranform

Page 3, Section 3.2: Is N the number of training examples and M the resolution?
Figure 2 not referred in the main text.






**Experience Assessment:**

I have published one or two papers in this area.

**Review Assessment: Checking Correctness Of Derivations And Theory:**

I assessed the sensibility of the derivations and theory.

**Review Assessment: Checking Correctness Of Experiments:**

I assessed the sensibility of the experiments.

**Review Assessment: Thoroughness In Paper Reading:**

I read the paper at least twice and used my best judgement in assessing the paper.

---

> ### Author Response · Authors · 2019-11-13
> **Response to review #2**
>
> Thank you for your review.
>
> “Being able to easily scale to higher resolutions is claimed to be one of the main advantages, but I am not convinced that this is useful under this setting. If I understand this correctly, the number of eigenvectors k is fixed, and projecting the SDF field to this space would remove the higher frequency components of the shape. So wouldn't the number of eigenvectors be the bottleneck in representational precision, not the resolution of the output space?”
>
> First of all we want to highlight that the number of eigenvectors k needed to capture the data variance are very small relative to the larger resolutions we consider. For example in our experiments with 128^3 resolution  k= 2048 for category ShapeNet-cars.  Plot of captured variance of category ShapeNet-cars for resolution 128^3 can be found in the new version in Appendix A1.
>
> Secondly, we emphasize that the reason the proposal is scalable is that it avoid having a 3D convolutional decoder which will be by construction much slower than just predicting k coefficients, even if k were big in practice, which it isn’t.
>
> "What is the chosen k (number of eigenvectors)? It says k was "chosen to capture 99.5% of the variance within the dataset", but I could not find how exactly it was chosen and what value of k was used (I apologize if I missed)."
>
> The chosen number of eigenvectors ranged from 512 to 2048. Some ShapeNet categories have small number of examples (such as phone, watercraft, or bench - approximately 1 000 examples) while others are substantially bigger (table, car, chair  - close to 8 000 examples). We have added more details regarding this in Appendix A1.
>
>
> "Also, I think the PCA is category-specific (page 4, section 4.1). Is k dependent on the category or is it the same across all categories?"
>
> Yes, as mentioned before, since ShapeNet categories differ in size (from ~1k phones to ~8k cars), our choice for number of eigenvectors differs as well.
>
> "Some of the other methods (if not all) used for comparison are not category-specific, so if this is true, I think the comparison may not be entirely fair and it should be made clearer."
>
> Among methods in Section 4.1, only 3D R2N2 explicitly train their network  jointly on all categories. However, our framework can be trivially generalized to a category-agnostic one. The only modification would be a larger number of eigenvectors.
>
> In Section 4.4, all methods were trained per category.
>
> "Minor typos"
>
> Thank you for noting the typos we have corrected them in the manuscript.
>
> "Figure 2 not referred in the main text."
>
> It is referred in the main text as “Figure 4.3”, this was latex referencing error and we have now corrected it.

---

### Author Response · Authors · 2019-11-13
**Changes in the manuscript**

Dear Reviewers, Thank you for your comments that help us to revise the manuscript. Based on the reviews we have made the following changes in the manuscript:

We have updated Equation (2) and revised Section 3 to improve clarity
We have added the reference and description of  main figure in section 3.2
We have fixed some typos in the Experimental section.
We have revised all minor grammatical and spelling errors noted by reviewers.
We added an Appendix which provides more details and analysis for both the choice of the number of coefficiencts and also discussed why other shape representations are not naturally combined with PCA

---

### Decision · Program_Chairs · 2019-12-19

**Decision:**

Reject

**Comment:**

This paper proposes to use PCS to replace the conventional decoder for 3D shape reconstruction. It shows competitive performance to the state of the art methods. While reviewer #3 is overall positive about this work, both reviewer #1 and #2 rated weak rejection. Reviewer #1 concerns that important details are missing, and the discussion of results is insufficient. Reviewer #3 has questions on the clarity of the presentation and comparison with SOTA methods. The authors provided response to the questions, but did not change the rating of the reviewers. The ACs agree that this work has merits. However, given the various concerns raised by the reviewers, this paper can not be accepted at its current state.